# Association between Visual Perception and Socioeconomic Status in Malaysian Preschool Children: Results from the Test of Visual Perceptual Skills-4

**DOI:** 10.3390/children10040749

**Published:** 2023-04-20

**Authors:** Mariah Asem, Sumithira Narayanasamy, Mahadir Ahmad, Masne Kadar, Mohd Izzuddin Hairol

**Affiliations:** 1Centre for Community Health Studies (ReaCH), Faculty of Health Sciences, Universiti Kebangsaan Malaysia, Jalan Raja Muda Abdul Aziz, Kuala Lumpur 50300, Malaysia; mariah.asem@yahoo.com (M.A.); sumithira@ukm.edu.my (S.N.); mahadir@ukm.edu.my (M.A.); 2Centre for Rehabilitation and Special Needs Studies (iCaRehab), Faculty of Health Sciences, Universiti Kebangsaan Malaysia, Jalan Raja Muda Abdul Aziz, Kuala Lumpur 50300, Malaysia; masne_kadar@ukm.edu.my

**Keywords:** preschool children, Test of Visual Perception Skills—4th edition (TVPS-4), visual perception, socioeconomic status

## Abstract

Visual perception in children can be evaluated using the Test of Visual Perceptual Skills—4th edition (TVPS-4) with normative data developed for the U.S. population. It is widely used by healthcare practitioners in Malaysia, despite reports that children in Asia outperform their U.S. peers in visual perception assessment. We compared TVPS-4 scores among 72 Malaysian preschool children (mean age: 5.06 ± 0.11 years) with U.S. norms and investigated the association between socioeconomic factors and TVPS-4 scores. Malaysian preschoolers had significantly higher standard scores (116.60 ± 7.16) than the U.S. norms (100 ± 15; *p* < 0.001). They also had significantly higher scaled scores (between 12.57 ± 2.10 and 13.89 ± 2.54) than the U.S. norms (10 ± 3, all *p* < 0.001) for all subtests. Multiple linear regression analyses revealed that socioeconomic variables were not significant predictors for five visual perception subtests and the overall standard score. The visual form constancy score could be predicted by ethnicity (β = −1.874, *p* = 0.03). The visual sequential memory score could be predicted by the father’s employment status (β = 2.399, *p* < 0.001), mother’s employment status (β = 1.303, *p* = 0.007), and low household income (β = −1.430, *p* < 0.037). In conclusion, Malaysian preschoolers outperformed their U.S. peers in all TVPS-4 subtests. Socioeconomic variables were associated with visual form constancy and visual sequential memory, but not with the other five subtests or TVPS-4’s overall standard scores.

## 1. Introduction

Visual perception, also known as visual processing or visual information processing, refers to the brain’s ability to receive, process, and interpret visual sensory information received by a person through their eyes [1]. It is a goal-directed process that requires attending to and identifying critical visual features in the environment, integrating the visual information with other sensory systems, and interpreting and attaching meaning to the information to engage with the environment. These processes of converting visual stimuli into meaningful information occur at a high level of executive function in our brains [2]. In everyday life, visual perception allows us to perform tasks such as recognizing geometric shapes and interpreting road signs.

Visual perception deficits in children can significantly impact their learning-related skills, such as reading, spelling, writing, and numeracy. Those with such deficits have been found to have lower academic achievement levels [3,4,5], poorer handwriting legibility [6], inferior development of social skills [7], lower self-esteem, and lower overall quality of life [2]. In addition, children’s development of perceptual and motor skills is reported to be affected by their culture [8,9], language [10], region [11], academic level [12,13], race, and socioeconomic status [14]. For example, children from higher socioeconomic backgrounds typically outperform those from middle and low socioeconomic backgrounds [14]. A recent study reported that preschoolers from low-income families had lower than average visual–motor integration performance [15] and were likelier to fail visual screening tests [16] than those from higher-income families. Thus, young children’s visual perceptual abilities must be correctly measured, as they are crucial to their functional task performance [17] and, consequently, their school readiness.

The Test of Visual Perceptual Skills—fourth edition (TVPS-4) is the latest update of one of the most widely used visual perceptual assessment tools in pediatric occupational therapy [18]. It evaluates seven visual perceptual subskills: visual discrimination (the ability to recognize details in visual images), visual memory (the ability to recall the object that is seen), visual–spatial relationships (the ability to identify an object located in space in relation to a reference object), visual form constancy (the ability to recognize a shape despite changes in size, orientation, direction, and distance), visual sequential memory (the ability to remember a series of objects arranged in a particular order), visual figure–ground (the ability to separate the elements of an image to perceive an object against a background), and visual closure (the ability to visualize a complete whole when only a partial image is presented) [18]. The test’s manual states that it is suitable to be administered to individuals between 5 and 21 years old, with the norms developed from a sample of English-speaking individuals residing in the United States.

Previous studies using various editions of the TVPS have reported conflicting findings regarding performance among children from various populations. The performances of Australian children (aged 6–12) matched well with those of their U.S. peers [19,20]. On the other hand, the performances of Chinese-speaking children in Hong Kong exceeded the U.S. standard norms [10,11,21]. In contrast, five-year-old South African children reported lower scores [22,23], while another South African study involving 6- to 11-year-old children found that their scores matched well with the U.S. norms [24]. These findings indicate that diverse cultural, linguistic, and socioeconomic groups might contribute to different TVPS-4 outcomes. Thus, standardized tests such as the TVPS-4 must be validated in particular cultural settings before the acquired results can be generalized [25].

To date, only one published study in Malaysia has measured TVPS performance in young children (eight-year-olds), and those with low academic achievement achieved lower scores than those with average academic achievement [13]. Currently, there are no published normative data on Malaysian preschool children’s TVPS scores, even though the test is widely used by pediatric practitioners throughout the country. Preschool age is critical for children in terms of developing their cognitive skills and school readiness. Thus, as visual perception is a process that involves both vision and cognition, the test should be conducted in this age group. Whether the socioeconomic background of Malaysian preschoolers affects their visual perception, as measured with the TVPS-4, remains to be investigated.

This study’s first objective was to compare the TVPS-4 scores of a sample of multi-ethnic Malaysian preschoolers with the U.S. standard norms. The second objective was to determine the relationship between the preschoolers’ TVPS-4 scores and their socioeconomic statuses, including ethnicity, household income, parent’s educational level, employment status, and preschool enrollment age.

## 2. Materials and Methods

The participants comprised a convenience sample of Malaysian preschool children aged between 5 and 6 years old and residing in the Klang Valley area. All participants had distance and near visual acuity of 0.3 logarithms of the minimum angle of resolution (logMAR) or better in each eye, and stereopsis of at least 150 s of arc or better as measured using the Frisby stereotest, to ensure that reduced vision did not affect their visual perception [26]. Other inclusion criteria included full-term birth history with no known physical and cognitive disabilities, as reported by the parents.

The sample size was calculated based on a pilot study of eight-year-old children using the Test of Visual Perceptual Skills—Revised (TVPS-R). The effect size f^2^ was estimated to be 0.29. Using G*Power 3.1 analysis for multiple linear regression [27], a significance level (α) of 5%, a power of 80%, and 7 socioeconomic predictor variables, the minimum required sample size was 63, taking 10% dropout into account. As a result, 55% of the participants were chosen from government preschools and 45% from private preschools, in order to reflect a nationally representative sample [28].

Ethics approval was obtained from the Research Ethics Committee, Universiti Kebangsaan Malaysia (UKM PPI/111/8/JEP-2021-472); the Ministry of Education, Malaysia (KPM.600-3/2/3-eras(11184); and the Department of Education, Federal Territory of Kuala Lumpur (JPWPKL.600-9/1/5 Jld.4(18)). The study participation invitation was emailed to 20 registered preschools (16 government and 4 private preschools) based on a list on the Ministry of Education website. Of the 20 preschools, 5 (3 government and 2 private preschools) responded to the invitation and agreed to participate in the study. Participants were recruited via convenience sampling; their homeroom teachers distributed the invitation forms, consent forms, and questionnaires to the parents/guardians. Those who agreed to their child participating then returned the completed forms before data collection commenced.

Participants who received consent from their parents/guardians and completed the sociodemographic questionnaire underwent a vision screening test to ensure that they complied with the inclusion criteria. The questionnaire included information regarding the child’s sex, age, ethnicity, date of birth, household income range, parents’ education level, parents’ employment status, preschool enrolment age, and birth and health history.

The TVPS-4’s seven subtests were administered to each participant using the prescribed multiple-choice response format. The subtests identified seven visual perception domains: visual discrimination, visual memory, visual–spatial relationships, visual form constancy, visual sequential memory, visual figure–ground, and visual closure [18]. Two demonstration plates were presented for each subtest, followed by eighteen test plates with increasing difficulty. Participants indicated their selected answers by responding verbally or through pointing gestures. The TVPS-4 required approximately 30–40 min, cumulatively, to administer. Mandatory break time was given between subtests to ensure that the participant did not lose focus or interest and to minimize fatigue. All subtests were conducted in the preschool’s classrooms early in the day to ensure sustained concentration. The researchers ensured that noise was minimized and no participants were disturbed by their peers during the administration of the tests. If the participants could not finish the tests within the same session, they were able to continue the next day.

The raw score was obtained for each subtest by adding the total number of correct responses. These raw scores were converted to scaled scores to allow for comparison with the test’s normative samples for each subtest. Next, a standard score was obtained from the sums of the scaled scores, reflecting the participant’s overall performance on the TVPS-4.

Statistical analyses were performed using the Statistical Package for the Social Sciences (SPSS) (version 26). Descriptive statistics, including mean and standard deviation (SD), were calculated for TVPS-4 scores based on age, sex, and socioeconomic variables (sex, ethnicity, preschool type, household income, father’s education level, mother’s education level, father’s occupation status, and mother’s occupation status). Frequencies and percentages were also calculated for all categorical variables. The normality of distributions and equality of variances were assessed to determine whether the assumptions for parametric statistics were met. When variables did not meet the assumptions for parametric tests, non-parametric statistical tests were employed.

An independent sample t-test was carried out to compare TVPS-4 performance between sexes. One-way analysis of variance (ANOVA) was used to compare TVPS-4 scores between ethnicities. One-sample t-tests were performed to compare the measured TVPS-4 scores with the U.S. normative score (mean = 100 and SD = 15) and the U.S. normative scaled score for each subtest (mean = 10 and SD = 3). The level of statistical significance was determined to be *p* < 0.05.

For categorical/ranked socioeconomic variables (ethnicity, preschool type, household income, father’s education, mother’s education, father’s occupation status, and mother’s occupation status), Kendall’s tau-b correlation analyses were used to determine their associations with TVPS-4 scores. Bonferroni corrections were conducted to correct for multiple testing, where the determined level of statistical significance (0.05) was divided by the number of tests (8 in total). Finally, multiple linear regression analyses with the Enter method were performed to determine how each socioeconomic factor could explain variances in TVPS-4 standard scores. All socioeconomic factors with a *p*-value < 0.15 from Kendall’s tau-b were included in the model. A higher significance level for variable selection was considered to ensure that variables relevant to the outcome were not missed and to avoid deleting less significant variables that may have had practical and clinical reasoning [29]. As the number of participants of Chinese, Indian, and other ethnicities was relatively small (18.1%), they were re-categorized as non-Malays for the correlation and regression analyses. Ethnicity was labeled as 0 (non-Malays) and 1 (Malays). Parental employment status was labeled 0 for self- or unemployed and 1 for formally employed. The self-employed were combined with the unemployed as they were mainly involved in the gig economy, thus without consistent, regular work or formal contributions to social security. The parental education level was labeled as 0 for low (up to secondary school) and 1 for high (tertiary education).

## 3. Results

A total of 72 preschool children participated in the study, with a mean age of 5.06 ± 0.11 years. The mean preschool enrolment age was 5.06 ± 0.95 years. The majority of the participants were of Malay ethnicity (81.9%). The participants’ socioeconomic data are presented in Table 1.

The mean TVPS-4 standard score for the sample was 116.60 ± 6.32. There was no significant difference in TVPS-4 standard score between boys (116.55 ± 7.78) and girls (116.67 ± 6.32) (t(70) = −0.069, *p* = 0.95). The mean standard score was highest among participants of Chinese ethnicity (122.75 ± 6.19) compared to those from other ethnic groups, but the difference was not statistically significant (F(3,68) = 1.27, *p* = 0.29). When the participants were regrouped as Malays and non-Malays, the mean standard score for non-Malays was higher than that for Malay participants, although not significantly different (119.46 ± 7.41; t(79) = −1.161, *p* = 0.112). The TVPS-4 standard scores according to sex and ethnicity are summarized in Table 2.

Table 3 compares the study participants’ standard and scaled scores for the seven TVPS-4 subtests with the U.S. normative scores. The study participants’ mean standard score (116.60 ± 7.16) was significantly higher than the U.S. normative score (100 ± 15) (t(71) = 19.68, *p* < 0.001, effect size = 2.32). The highest score was obtained for the visual closure subtest (13.89 ± 2.54), and the lowest for the figure–ground subtest (12.57 ± 2.10). For all seven subtests, the study participant’s scores were significantly higher than the U.S. normative scores (all *p* < 0.001, effect sizes between 0.92 and 1.61).

Table 4 presents the values of Kendall’s tau-B, *τ*_b_, for the TVPS-4 standard score and the scores for its seven subtests. Except for the visual closure subtest, at least one socioeconomic variable showed a significant relationship at *p* < 0.15 with the TVPS-4 scores.

With the significant socioeconomic variables included in the multiple regression analyses, the regression models could not significantly predict the participants’ TVPS-4 standard scores (F(5,66) = 1.28, *p* = 0.28) nor the scaled scores for visual discrimination (F(3,68) = 2.152, *p* = 0.102), visual memory (F(1,70) = 2.883, *p* = 0.094), visual–spatial relation (F(2,69) = 0.111), and visual–ground (F(1,70) = 2.209, *p* = 0.142) subtests.

The visual form constancy scores could be predicted significantly with a regression model (R^2^ = 0.065, F(1,70) = 4.861, *p* = 0.031) according to ethnicity (β = −1.874, *p* = 0.03); it was predicted to be lower for children with Malay ethnicity.

The visual sequential memory score could also be predicted significantly by a multiple linear regression model (R^2^ = 0.321, F(5,66) = 6.266, *p* < 0.001) according to the mother’s employment status (β = 1.303, *p* = 0.007), father’s employment status (β = 2.399, *p* < 0.001), and low household income (β = −1.430, *p* < 0.037). The visual sequential memory score was predicted to be lowest if the child had both a mother and father who were unemployed and was from a low-income household. Table 5 summarizes the multiple regression analyses for the two subtests.

## 4. Discussion

This cross-sectional study measured visual perception in a sample of Malaysian preschool children based on the TVPS-4 scores and investigated whether certain sociodemographic factors were predictive of their visual perception performance. The results indicated that Malaysian preschool children outperformed their US-referenced peers on the TVPS-4. There was also a trend where children of Chinese ethnicity in the current study had higher TVPS-4 standard scores than other ethnic groups. These findings are similar to those reported for Hong Kong Chinese preschoolers, who scored significantly higher than English-speaking U.S. children [10,11]. In contrast, a study involving five-year-old South African children reported lower scores [22,23]. These conflicting findings suggest that different countries with different cultural or socioeconomic backgrounds could have different TVPS performances. Thus, healthcare practitioners must consider these variations when using TVPS-4 as part of their pediatric clinical evaluations.

One reason that could explain the difference in visual perception performance between the Malaysian sample and the U.S. norms is the difference in the preschool education goals between the two countries. The learning standards that Malaysian preschool students must master include recognizing the alphabet, reading and understanding reading materials, and copying words and phrases [30]. Therefore, Malaysian preschool children are expected to be able to read and write letters as part of their academic tasks before they enter primary school. Similarly, reading literacy is heavily promoted in Hong Kong’s curriculum [31], with substantial literacy attainment among Hong Kong children by the age of four [32]. Compared to preschool education in Malaysia, where the preschool curriculum is academically oriented and teacher-centered, U.S. education is more child-centered, with a play-based approach that emphasizes children’s individualism, independence, creativity, and freedom [33]. Teachers encourage children’s self-expression and creativity, and more time is spent playing according to their interests [34,35]. Therefore, exposure to these academic-oriented activities due to the differences in school curriculum systems and their different goals or approaches may have contributed to the higher TVPS scores achieved by Malaysian preschoolers compared to the published norms.

In this study, the visual form constancy scores were predicted to be lower for children of Malay ethnicity compared to non-Malay children residing within the same area in Malaysia. Specifically for the visual form constancy subtest, previous studies reported that the scores were lower for South African children than the U.S. norms [36], while Hong Kong preschool children had higher scores [21]. Thus, differences in scores on the individual subtests occur not only in children from different populations, but also for multi-ethnic children in a single population. In TVPS-4, visual form constancy is measured by asking the participants to match two similar images, one of which could be bigger, smaller, or rotated. An earlier study on Malaysian preschoolers reported that Malay children were more likely to fail vision screening, and non-Malay children (including those of Chinese ethnicity) were likelier to have better visual acuity and stereopsis [16]. Both tasks involved object recognition and visual matching, which could be related to Malay children’s relatively lower visual form constancy performance. In addition, another study reported that visual skills, including visual form constancy, were significant predictors for successfully reading Chinese characters [37]. It is possible that children from some cultural backgrounds would more easily recognize some of the figures and shapes presented in TVPS-4. In addition, the intricate geometrical and orthographical design of Chinese characters could help children to develop their spatial orientation skills [38]. Thus, we speculate that differences in ethnicity and cultural background could affect how children perform on the visual form constancy task. Although the number of children of Chinese ethnicity in the current study was relatively small, the findings support the results from these previous studies.

The regression model revealed that the visual sequential memory score was be lowest for children whose parents were unemployed and had a low household income, which could indicate relatively lower socioeconomic status. Sequential memory is positively associated with reading and comprehension [39], as well as numeracy scores [40]. Thus, the lower sequential memory scores in these children are of concern, as parental unemployment, for fathers in particular, has a negative impact on children’s educational attainment [41]. Although the scores for the study participants were still relatively higher than the U.S. norms, this could be an early indicator that a gap in educational attainment for children within the same population may exist due to their parents’ employment statuses. Parents who are not employed may have lower incomes, and, thus, may struggle to create a conducive learning environment. Indeed, children who grow up in lower-income families are found to be less prepared when entering preschool, resulting in lower academic performance when they enter school [42,43]. Previous studies published in this country have reported that socioeconomic status is significantly associated with various developmental aspects, such as cognitive level [44,45] and visual–motor skills [15]. Indeed, parental employment and their financial involvement in their children’s learning were significantly associated with children’s cognitive levels [46], reading interests [47], and academic achievements [48], contributing positively to children’s development.

A recent study reported that children from higher-income families had better working memory skills during their early school years than their peers from low-income families [49]. Although the TVPS-4 subtest scores for this study’s sample had significant socioeconomic predictors, they were still higher than the test’s norms. These higher scores were possibly related to the study participants residing in Klang Valley, an area with the country’s highest gross domestic product (GDP) per capita. Children from areas with lower GDPs per capita would likely achieve lower sequential memory scores, and, perhaps, lower scores for the other visual perception measures in the TVPS-4.

This study has several limitations. Firstly, it is based on a population sample from only one district in the Klang Valley, a metropolitan and developed urban area in Malaysia, and thus may not represent the entire Malaysian population. The results of this study may not be generalized to children in rural areas with relatively lower socioeconomic conditions. In addition, the majority of the children were Malays, with smaller numbers of children from other ethnicities. Therefore, future studies need to expand the sampling of children in other areas, especially rural areas and those belonging to ethnic minority groups. In addition, the socioeconomic measures were categorical variables; continuous measures may be more sensitive to detecting the magnitude of the relationship between socioeconomic variables and TVPS-4 scores. Lastly, some participants took two sessions over two consecutive days to complete all subtests, while others were able to complete it in a single session. Therefore, different amounts of time taken to complete the tests could have affected the scores.

## 5. Conclusions

Malaysian preschoolers scored significantly higher than the U.S. norms for all visual perception parameters measured using the TVPS-4, suggesting that standardized test norms may not necessarily reflect the performance of a population outside of where they were developed. The associations between socioeconomic variables were not significant for five out of TVPS-4’s seven subtest scores, nor for the standard score. The form constancy score could be predicted by ethnicity, while the sequential memory score could be predicted by parents’ education level and household income. Therefore, healthcare practitioners must consider these variables when using TVPS-4 as part of the clinical evaluation of children’s visual perception.

## Figures and Tables

**Table 1 children-10-00749-t001:** Socioeconomic data of the study participants.

Variable		Number of Participants, n (%)
Sex	Male	42 (58.3%)
	Female	30 (41.7%)
Ethnicity	Malay	59 (81.9%)
	Chinese	4 (5.6%)
	Indian	4 (5.6%)
	Other	5 (6.9%)
Preschool type	Government	42 (58.3%)
	Private	30 (41.7%)
Household income	Low (<MYR4850)	34 (47.2%)
	Middle (MYR4851 to MYR10959)	25 (34.7%)
	High (>MYR10959)	13 (18.1%)
Father’s education level	Up to secondary school	30 (41.7%)
	Tertiary education	42 (58.3%)
Mother’s education level	Up to secondary school	23 (31.9%)
	Tertiary education	49 (68.1%)
Father’s employment status	Formally employed	61 (84.7%)
	Self-/unemployed	11 (15.3%)
Mother’s employment status	Formally employed	42 (58.3%)
	Self-/unemployed	30 (41.7%)

**Table 2 children-10-00749-t002:** TVPS-4 standard scores (mean ± standard deviation) based on participants’ sex and ethnicity.

Variable		TVPS-4 Standard Score	*p*-Value
Sex	Male	116.55 ± 7.78	0.95
	Female	116.67 ± 6.32	
Ethnicity	Malay	115.97 ± 7.01	0.29
	Chinese	122.75 ± 6.19	
	Indian	118.00 ± 8.45	
	Others	118.00 ± 8.19	

**Table 3 children-10-00749-t003:** Comparison between the study participants and the U.S. norms for TVPS-4 standard score (normative mean: 100 ± 15) and scaled scores (normative mean: 10 ± 3).

TVPS-4 Score	Study Participants	U.S. Norms	95% Confidence Interval ^1^		*t*-Value	*p*-Value	Cohen’s *d*
			Lower	Upper			
Standard score	116.60 ± 7.16	100 ± 15	14.92	18.28	19.68	<0.001	2.32
Visual discrimination	13.68 ± 2.45	10 ± 3	3.07	4.24	12.50	<0.001	1.47
Visual memory	13.38 ± 2.22	10 ± 3	2.85	3.90	12.92	<0.001	0.92
Visual–spatial relations	12.90 ± 2.33	10 ± 3	2.36	3.45	10.59	<0.001	1.25
Visual form constancy	13.54 ± 2.85	10 ± 3	2.87	4.21	10.55	<0.001	1.24
Visual sequential memory	13.67 ± 2.19	10 ± 3	3.15	4.18	14.22	<0.001	1.61
Visual figure–ground	12.57 ± 2.10	10 ± 3	2.08	3.06	10.41	<0.001	1.23
Visual closure	13.89 ± 2.54	10 ± 3	3.29	4.49	13.01	<0.001	1.03

^1^ The 95% confidence interval refers to the mean differences between the U.S. norms and the sample means.

**Table 4 children-10-00749-t004:** Kendall’s tau-B analyses for the association between TVPS-4 standard and scaled scores, as well as socioeconomic variables.

Variable	Visual Discrimination		Visual Memory		Visual–Spatial Relation		Visual Form Constancy		Visual Sequential Memory		Visual Figure–Ground		Visual Closure		TVPS-4 Standard Score	
	*τ* _b_	*p*-Value	*τ* _b_	*p*-Value	*τ* _b_	*p*-Value	*τ* _b_	*p*-Value	*τ* _b_	*p*-value	*τ* _b_	*p*-Value	*τ* _b_	*p*-Value	*τ* _b_	*p*-Value
Ethnicity	0.046	0.657	0.161	0.119 *	0.155	0.13 *	0.209	0.041 *	0.042	0.684	0.059	0.551	0.036	0.729	0.155	0.120 *
Preschool type	0.141	0.168	0.107	0.302	0.038	0.712	0.129	0.204	0.20	0.844	0.159	0.124 *	−0.010	0.922	0.166	0.096 *
Household income	0.196	0.044 *	0.111	0.259	0.076	0.438	0.019	0.348	0.215	0.029 *	0.124	0.231	0.076	0.439	0.181	0.056 *
Mother’s education level	0.163	0.113 *	0.100	0.333	0.186	0.07 *	0.046	0.649	−0.002	0.985	0.068	0.513	−0.020	0.846	0.115	0.246
Father’s education level	0.053	0.604	0.059	0.566	0.121	0.237	0.097	0.343	0.165	0.110 *	0.135	0.192	0.104	0.311	0.169	0.089
Father’s employment status	−0.071	0.487	−0.013	0.902	−0.31	0.76	−0.032	0.755	−0.345	0.001 ^†,^*	0.049	0.635	0.002	0.988	−0.102	0.307
Father’s employment status	0.021	0.84	0.023	0.821	0.143	0.163	0.068	0.501	−0.294	0.004 ^†,^*	−0.059	0.571	−0.003	0.977	0.006	0.950

^†^ Significant after Bonferroni correction for multiple testing, *p* < 0.00625; * significant at *p* < 0.15.

**Table 5 children-10-00749-t005:** Multiple regression analyses for the visual form constancy and visual sequential memory subtests.

Subtest	Predictors	Unstandardized Coefficient, *B*	Standard Error	95% Confidence Interval		*p*-Value
Visual form constancy	Ethnicity	−1.874	0.85	−3.57	−0.18	0.03
Visual sequential memory	Mother’s employment status	1.303	0.467	0.370	2.236	0.007 *
	Father’s employment status	2.399	0.599	1.204	3.59	<0.001 *
	Father’s education level	0.198	0.527	−0.854	1.251	0.708
	Household income (low)	−1.430	0.672	−2.772	−0.90	0.037 *
	Household income (middle)	−0.817	0.643	−2.09	−0.47	0.209

* significant at *p* < 0.05.

## Data Availability

The data presented in this study are openly available in Figshare at https://doi.org/10.6084/m9.figshare.22147172, accessed on 12 April 2023.

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
