# Peer review of "Association between Visual Perception and Socioeconomic Status in Malaysian Preschool Children: Results from the Test of Visual Perceptual Skills-4"

_children, 2023, doi:10.3390/children10040749_

Round 1

Reviewer 1 Report

The manuscript deals with a very relevant topic and research questions are interesting. The author M. Asem and colleagues compared Test of Visual Perceptual Skills-4 (TVPS-4) scores in a Malaysian cohort with U.S. standard norms and investigated socioeconomic status to predict TVPS-4 scores. They report higher standard scores in Malaysian preschooler at the age of 5 years than in U.S. norms. Socioeconomic variables were not significant predictors for five visual perception scores and overall standard score. The authors conclude that Malaysian preschoolers perfomed better than their U.S: peers and socioeconomic variables were partly related to TVPS-4 subtests.

The research questions of the study are interesting, the participants and methods are well selected. Only the selection of the many statistical tests in a small sample without multiple-comparison correction is not appropriate. The results are reported in a clear, comprehensible, and understandable manner. I recommend a minor revision.

Comments:

1.   Abstract: Please add the sample size.

2.      Line 41 behind writing a comma is missing

3.      Line 50 to 52: Please add a reference.

4.      Line 55 to 62: Please add a reference for the definition of visual perceptual skills.

5.      Line 124 to 125: Please format the reference according to the journal guidelines.

6.      Line 138: Please check “TVPS-4” for consistent usage throughout the manuscript.

7.      Please limit the statistical analyses. Calculate only the prediction (multiple regression analysis) of each variable of Socioeconomic Status on the TVPS-4 standard score and not on the subscores.

8.      Please perform a correction for multiple testing.

9.      Table 1: The title is missing. The total number of participants for “father’s employment status” is n = 42. Thee is a discrepancy with the total number of participants in the cohort n = 72.

10.   Table 2: The table could be in the appendix.

11.   Table 3: Line to (visual discrimination) standard deviation has 3 digits after the decimal point. In the other line are only 2 digits after the decimal point.

12.   Table 3: Add notes below the table and explain what the 95% confidence interval refers to.

13.   Table 4: The title is missing. I would like to point out that far too many tests were performed for the small sample size. Please limit the regression analysis to the TVPS-4 standard score.

14.   Please sensitively articulate the differences in the Education in Malaysia and U.S. for example western ideology.

15.   Line 248 to 249: Please avoid repetition.

16.   References: 14: Please check if the capitalization is correct here.

Author Response

Many thanks to Reviewer 1 for the insightful comments, which have improved the manuscript significantly.

Response to Reviewer 1

Thank you, Reviewer 1, for the comments. Below are our point-by-point responses.

  1. Abstract: Please add the sample size.

Our response: sample size has been added to the abstract. It now reads (line 14):

“We compared TVPS-4 scores among 72 Malaysian preschool children (mean age: 5.06±0.11 years) with U.S. norms and investigated the association between socioeconomic factors and TVPS-4 scores.”

  1. Line 41 behind writing a comma is missing.

Our response: a comma has been added (line 14).

  1. Line 50 to 52: Please add a reference.

Our response: A reference (number 17) has been added (line 51):

Brown, G.T.; Rodger, S.; Davis, A. Test of Visual Perceptual Skills - Revised: An Overview and Critique. Scand J Occup Ther 2003, 10, 3-15.

  1. Line 55 to 62: Please add a reference for the definition of visual perceptual skills.

Our response: A reference has been added (line 63):

Martin, N.A. Test of Visual Perceptual Skills (TVPS-4); 3rd ed.; Academic Therapy Publications: Novato, CA, 2017

  1. Line 124 to 125: Please format the reference according to the journal guidelines.

The reference has been formatted according to the journal guidelines.

  1. Line 138: Please check “TVPS-4” for consistent usage throughout the manuscript.

Our response: we have checked throughout the manuscript, and “TVPS-4” is now consistent.

  1. Please limit the statistical analyses. Calculate only the prediction (multiple regression analysis) of each variable of Socioeconomic Status on the TVPS-4 standard score and not on the subscores.

Our response: in our submitted manuscript, we have already reported that the TVPS-4 standard score could not be predicted by the socioeconomic variables in the multiple linear regression analyses.

However, we found that scores of two subtests, visual form constancy and visual sequential memory, could be predicted partly with the multiple linear regression analyses. Therefore, these analyses on the subtests are crucial to be part of the Discussion.

Thus, we believe that the analyses on the subtest scores should remain in the paper, and Reviewers 2 & 3 also did not suggest they should be excluded.

  1. Please perform a correction for multiple testing.

Our response: We agree with Reviewer 1 that a correction for multiple testing should be conducted. In the statistical analysis part in Methods, we have added this sentence (line 157):

“Bonferroni corrections were conducted to correct for multiple testing, where the determined level of statistical significance (0.05) was divided with the number of tests (eight in total).”

These corrections are reflected in Table 4, where the significant results with corrected p-values (0.00625) are marked with along with a footnote.

  1. Table 1: The title is missing. The total number of participants for “father’s employment status” is n = 42. There is a discrepancy with the total number of participants in the cohort n = 72

Our response: The title for Table 1 is “Table 1: Socioeconomic data of the study participants.”

There are two categories of father’s employment status: formally employed (n=61) and self-/unemployed (n=11), making the total number 72. We have highlighted these numbers in the revised manuscript.  

  1. Table 2: The table could be in the appendix.

Our response: We thank Reviewer 1 for the suggestion. However, we believe Table 2 should remain in the main text to directly report the TVPS-4 scores between the sex and ethnicity of the participants.  

  1. Table 3: Line to (visual discrimination) standard deviation has 3 digits after the decimal point. In the other line are only 2 digits after the decimal point.

Our response: Thank you for pointing this out. It has now been corrected in Table 3 (visual discrimination: 13.68±2.45).

  1. Table 3: Add notes below the table and explain what the 95% confidence interval refers to.

Our response: a footnote has been added below Table 3, which reads:

1The 95% confidence interval refers to the mean differences between the U.S. norms and the sample means”

  1. Table 4: The title is missing. I would like to point out that far too many tests were performed for the small sample size. Please limit the regression analysis to the TVPS-4 standard score.

The title Table 4 has been included, which is:

“Table 4: Kendal Tau B analyses of the association between TVPS-4 standard & scaled scores and socioeconomic variables”

We believe that the analyses for the subtests are also necessary, as analyzing the standard score alone did not reveal any significant associations with the socioeconomic variables. Analyses of the subtests, on the other hand, revealed that the two components that made up the standard score (i.e. two out of seven subtests) were associated with socioeconomic variables.

Thus, each subtest should be analyzed separately. Relying on the standard score alone may cause us to miss any differences in visual performance that correspond to the subtests, which have implications in clinical practice when practitioners use the TVPS-4 to measure visual performance in young children. Therefore, we feel that Table 4 should remain in the manuscript. 

  1. Please sensitively articulate the differences in the Education in Malaysia and U.S. for example western ideology.

Our response: yes, we agree with Reviewer 1 that the differences should have been articulated more sensitively. We have omitted the phrase ‘western ideology’ from the sentence.

  1. Line 248 to 249: Please avoid repetition.

In the first paragraph of the Discussion, comparisons were made on the TVPS standard score between children from different populations. However, in the third paragraph, the comparisons were made specifically for the TVPS-4 visual form constancy. Therefore, the sentence in line 253 has been rephrased. It now reads:

“Specifically for the visual form constancy subtest, previous studies have reported that the scores were lower for South African children than the U.S. norms [40], while those from Hong Kong had higher scores [21].”

  1. References: 14: Please check if the capitalization is correct here.

Our response: the reference has been corrected.

Reviewer 2 Report

This is an interesting work looking into cultural differences in perception. While the topic is important and such projects should be encouraged, I am not sure the current results add much insight. Regarding the comparison with the US population, the main limitations are well recognized by the authors, as described in the discussion: “This study is based on a population sample from only one district in the Klang Valley, a metropolitan and developed urban area in Malaysia, thus may not represent the entire 293 population of Malaysia. The results of this study may not be generalized to children in rural areas with relatively lower socioeconomic conditions. Therefore, future studies need to expand the sampling of children in other areas, especially rural areas, to determine their visual perception level. In addition, the socioeconomic measures were categorical variables, whereas continuous measures may be more sensitive to detecting the magnitude of 298 the relationship between socioeconomic variables and TVPS-4 scores.”

In addition, the ethnicity variable is not too useful since there only a few non-Malaysian children tested (4+4+5 out of 72).

Maybe the way out is by a comparison with a suitable US sub-group, if possible. As noted by the authors, the educational systems differ between the two countries having different curricula. It seems that the Malaysian system has an advantage

Author Response

Response to Reviewer 2

Point 1:

This is an interesting work looking into cultural differences in perception. While the topic is important and such projects should be encouraged, I am not sure the current results add much insight.

Our response: We thank Reviewer 2 for their interest in our work on cultural differences in perception. We believe that the current results have implications in two ways. First, they add insights into the possible differences in visual processing mechanisms for children that may be affected by their cultural and socioeconomic differences. Secondly, they are significant for clinicians who use TVPS-4 as part of visual perception assessment in young children. Malaysian clinicians should be cautious about relying on TVPS-4 interpretation based solely on the US norms. This is to prevent overestimating Malaysian children’s abilities, which could cause delays in providing intervention for those who appear to be ‘borderline normal based on the US norms.

Point 2:

In addition, the ethnicity variable is not too useful since there only a few non-Malaysian children tested (4+4+5 out of 72). Maybe the way out is by a comparison with a suitable US sub-group, if possible. As noted by the authors, the educational systems differ between the two countries having different curricula. It seems that the Malaysian system has an advantage.

Our response: Yes, we agree that the ethnicity distribution of our sample did not represent the actual national distribution. We have addressed this issue in Limitations:

“The results of this study may not be generalized to children in rural areas with relatively lower socioeconomic conditions. In addition, the majority of the children were Malays, with smaller numbers of children from other ethnicities.”

Although the TVPS-4 manual states that the norms were developed from samples across different ethnicities, the majority were White/Caucasian (78.27%) and the data were not analyzed based on ethnicity. Two studies in South Africa reported that the TVPS in its current form may not be suitable for children with diverse cultural, language, and socioeconomic differences (Visser et al. 2017, Harris 2017). Although the number of children of ethnic minorities in our study was relatively small, our findings agree with the results from these previous studies. 

As Reviewer 2 pointed out, it seems that the Malaysian system has an advantage. Previous studies, as cited in the manuscript, have speculated that the education systems in Hong Kong could also influence the higher TVPS-4 scores obtained by Hong Kong children. Multiple environmental factors could influence the development of visual perceptual skills, which would be an interesting future research direction.

Reviewer 3 Report

The rationale of this study is clear, and it is indeed important to understand whether the TVPS-4 is valid for use in cultures other than those for whom it was originally developed. The present study is therefore important.

The study was conducted in government and private preschools. It should be noted that children of the same age at home would be another demographic. It would be good to know what proportion of children in the country spend their preschool years at home versus in preschool nurseries, and thus to what extent the present results reflect the situation for most children in Malaysia. For example, the results show slight but insignificant differences between scores in children of different ethnicities, and some significant associations with household income and other demographic factors. It would be worth acknowledging (in limitations) that not all demographic groups were explored here.

The relatively low scores in Malay students may reflect the ability of other children to relate to the presentation of the tasks. For example, the TVPS presents images and words, and it could be that children from some cultural backgrounds would more easily recognize some of the images. The authors should consider and at least briefly discuss this possibility.

In Methods, the inclusion/exclusion criteria do not seem to be complete. It is unclear whether children with known developmental or general health abnormalities were excluded. In addition, the age range associated with ‘preschool’ should be defined in the inclusion criteria.

The questionnaires were administered in classrooms which may be noisy environments with distractions. This may have affected the scores in at least some of the children, and perhaps it affected different demographic groups differently (if some tended to be in classes with more attentive children). The authors should address this briefly in Methods and discuss in the Discussion.

The tests were administered early in the day, with breaks between each of the subtests. It is unclear whether it was feasible to complete all tests on one morning – the authors should explain whether this was the case or students were brought back to complete tests.

The manuscript is generally well written but in places the writing needs attention. For example, ‘To date, only one published study in Malaysia measured TVPS performance in 76 young (children eight-year-olds)’ – the bracket should begin after ‘children’. This type of error occurs particularly in the Introduction, which the authors should check carefully before resubmitting.

Author Response

Response to Reviewer 3

Point 1: The study was conducted in government and private preschools. It should be noted that children of the same age at home would be another demographic. It would be good to know what proportion of children in the country spend their preschool years at home versus in preschool nurseries, and thus to what extent the present results reflect the situation for most children in Malaysia.

Our response: In Malaysia, enrolment into preschools is not compulsory. Therefore, it is hard to determine how many children undergo homeschooling during their preschool years. However, the enrolment rate for the year 2021 is 83.65% for five-year-olds (2021 Annual Report for the Malaysia Education Development Plan 2013-2025, accessible via https://www.padu.edu.my/wp-content/uploads/2022/09/PPPM_LT2021_BM_Final_compressed.pdf).

As quite a high number of 5-year-olds are formally enrolled in preschool, we believe our data could reflect the situation for most children in Malaysia.  

Point 2: The results show slight but insignificant differences between scores in children of different ethnicities and some significant associations with household income and other demographic factors. It would be worth acknowledging (in limitations) that not all demographic groups were explored here

Our response: We agree that the children recruited in our study did not fully represent the ethnic distribution in Malaysia. We have acknowledged this issue in the limitations, which now reads (line 305):

“The results of this study may not be generalized to children in rural areas with relatively lower socioeconomic conditions. In addition, the majority of the children were Malays, with smaller numbers of children from other ethnicities.”

Point 3: The relatively low scores in Malay students may reflect the ability of other children to relate to the presentation of the tasks. For example, the TVPS presents images and words, and it could be that children from some cultural backgrounds would more easily recognize some of the images. The authors should consider and at least briefly discuss this possibility.

Our response: We have expanded the Discussion on the children from some cultural backgrounds would more easily recognize some of the images presented in TVPS-4. In the Discussion, we have added the following (line 267):

“It is possible that children from some cultural backgrounds would more easily recognize some of the figures and shapes presented in TVPS-4. For example, Chinese characters’ intricate geometrical and orthographical design could help children develop their spatial orientation (Chen & Kao 2002).”

Point 4: In Methods, the inclusion/exclusion criteria do not seem to be complete. It is unclear whether children with known developmental or general health abnormalities were excluded. In addition, the age range associated with ‘preschool’ should be defined in the inclusion criteria.

Our response: In the manuscript, we have included the inclusion criteria (line 91):

“Participants were a convenience sample of Malaysian preschool children aged between 5 to 6 years old residing in the Klang Valley area”,

and in line 96:

“Other inclusion criteria include full-term birth history with no known physical and cognitive disability as reported by the parents.”

Point 5: The questionnaires were administered in classrooms which may be noisy environments with distractions. This may have affected the scores in at least some of the children, and perhaps it affected different demographic groups differently (if some tended to be in classes with more attentive children). The authors should address this briefly in Methods and discuss in the Discussion.

Our response:

In Methods, we have further elaborated on how the tests were administered (line 133):  

“All subtests were conducted in the preschool’s classrooms and early in the day to ensure sustained concentration. The researchers ensured that noise was minimized and each participant was not disturbed by their peers during the administration of the tests. If the participants could not finish the tests within the same session, they could continue the next day.”

We have also added some points in Limitations (line 311):

“Lastly, some participants took two sessions over two consecutive days to complete all subtests, while others could complete it in a single session. Therefore, different duration taken to complete the tests could have affected the scores.” 

Point 6: To date, only one published study in Malaysia measured TVPS performance in 76 young (children eight-year-olds)’ – the bracket should begin after ‘children’. 

Our response: We have corrected the sentence as suggested.